# Screening and Research on Skin Barrier Damage Protective Efficacy of Different Mannosylerythritol Lipids

**DOI:** 10.3390/molecules27144648

**Published:** 2022-07-21

**Authors:** Chenxu Jing, Jiling Guo, Zhenzhuo Li, Xiaohao Xu, Jing Wang, Lu Zhai, Jianzeng Liu, Guang Sun, Fei Wang, Yangfen Xu, Zhaolian Li, Daqing Zhao, Rui Jiang, Liwei Sun

**Affiliations:** 1Research Center of Traditional Chinese Medicine, The Affiliated Hospital to Changchun University of Chinese Medicine, Changchun 130021, China; jingchenxu77@sohu.com (C.J.); zhenzhuoli0705@163.com (Z.L.); xiaohaoclient@hotmail.com (X.X.); wj147427@163.com (J.W.); hildazhai@icloud.com (L.Z.); sunguang912314@163.com (G.S.); 18686561416@163.com (F.W.); 2Jilin Ginseng Academy, Changchun University of Chinese Medicine, Changchun 130117, China; janie131419@126.com (J.G.); liujz_1988@163.com (J.L.); zhaodaqing1963@163.com (D.Z.); 3Modern Hanfang Technology Company Limited, Guangzhou 510550, China; 13416399426@163.com (Y.X.); zachary1977@gmail.com (Z.L.)

**Keywords:** MEL-B, skin barrier damage, HaCaT cells, FLG, LOR

## Abstract

Mannosylerythritol lipids (MELs) may prevent skin barrier damage, although their protective mechanisms and active monomeric constituents remain unclear. Here, three MELs were extracted from *Candida antarctica* cultures containing fermented olive oil then purified using silica gel-based column chromatography and semipreparative HPLC. All three compounds (MEL-A, MEL-B, MEL-C) were well separated and stable, and reliable materials were used for NMR and HRESIMS chemical structure determinations and for assessing MELs’ protective effects against skin damage. Notably, MEL-B and MEL-C effectively protected HaCaT cells from UVB-induced damage by upregulating the contents of filaggrin (FLG) and transglutaminase-1 (TGM1), as determined via ELISA. Moreover, MEL-B treatment (20 μg/mL) of UVB-irradiated HaCaT cells led to the upregulation of both the expression of mRNA genes and the key proteins FLG, LOR, and TGM1, which are known to be decreased in damaged skin cells. Additionally, histopathological analysis results revealed a markedly reduced intracellular vacuolation and cell damage, reflecting improved skin function after MEL-B treatment. Furthermore, immunofluorescence results revealed that MEL-B protected EpiKutis^®^ three-dimensional cultured human skin cells from sodium dodecyl sulfate-induced damage by up-regulating FLG, LOR, and TGM1 expression. Accordingly, MELs’ protection against skin barrier damage depended on MEL-B monomeric constituent activities, thus highlighting their promise as beneficial ingredients for use in skin-care products.

## 1. Introduction

The skin barrier, which constitutes the major physical barrier between the human body and the external environment, acts to prevent external substances from entering the body to harm tissues and stimulate the immune response [1]. Skin is mainly composed of the epidermis, a barrier that prevents entry into the body of pathogens, irritants, and UV radiation, as well as the loss of water and solutes to help maintain homeostasis. Skin barrier integrity depends on the proper functioning of a specialized, stratified structural protein complex known as the cornified envelope (CE), which is a protein-keratinized capsule exhibiting an extremely stable layered structure [2]. Proper CE function depends on interactions among transglutaminase-crosslinked proteins, such as filaggrin (FLG) and loricrin (LOR), which are associated with skin barrier damage manifesting as dryness and desquamation. Crosslinking of both proteins by the endoenzyme transglutaminase-1 (TGM1) is vitally important for CE generation and maintenance [3]. In fact, numerous studies have demonstrated that losses of FLG, LOR, and TGM1 proteins impair epidermal formation and alter epidermal barrier function. In turn, such changes lead to differentiation disorders that adversely affect cell wall cutin formation that can compromise skin integrity and cause excessive skin dryness and desquamation, lipid-forming variations, skin allergies, and so on [4,5,6,7,8].

Mannosylerythritol lipids (MELs, Figure 1) are biosurfactants and surface-active biomolecules that possess excellent emulsifying activities, as well as beneficial pharmacological properties. Advantages of biosurfactants include antibacterial activities, low toxicity, high biodegradability, widespread applicability, and good performance under extreme conditions of temperature, pH, and salinity [9,10,11,12]. For instance, MELs have been shown to improve the appearance of damaged skin by reducing melanin production to alleviate undesirable dark spots by whitening the skin [13]. Moreover, several studies have demonstrated that MEL-B treatment can restore the skin’s ability to retain moisture [14,15]. However, very few studies have clarified whether MELs could protect the skin barrier from damage caused by external insults, such as UVB irradiation, prompting this study.

In order to broaden the applicability of MELs to cosmetics and pharmaceuticals, here we focused our attention on MELs’ effects on skin barrier damage. All MEL compounds (MEL-A, MEL-B, and MEL-C) were successfully separated using silica gel-based column chromatography and further purified using semipreparative HPLC in order to obtain stable and reliable material for use in biological activity determinations. Subsequently, MEL-B effects were evaluated using in vitro models, including a UVB-irradiated HaCaT skin cell model and a sodium dodecyl sulfate (SDS)-induced EpiKutis^®^ three-dimensional (3D) human skin cell model. Notably, our results demonstrated for the first time that MEL-B treatment repaired skin barrier damage by up-regulating the expression of key skin cell mRNA genes and the proteins FLG, LOR, and TGM1. These results provide a theoretical basis for identifying MELs’ active components and for determining the mechanism underlying MEL-B alleviation of skin barrier damage.

## 2. Results

### 2.1. Isolation and TLC Verification of MELs

Three spots (MEL-A, MEL-B, and MEL-C) in a crude MEL extract were detected using TLC that had *R_f_* values of 0.7, 0.4, and 0.2, respectively (data not shown). To obtain purified MEL preparations, the elution rate was controlled at 0.8 mL/min while a 20 mL volume of eluent was collected in each test tube and spotted onto a TLC plate. All fractions with similar *R_f_* values were pooled and used for subsequent experiments.

### 2.2. Quantitative Detection of MELs

MELs’ monomeric constituents were confirmed using a high-performance liquid chromatography-evaporative light scattering detector (HPLC-ELSD) system equipped with a low-temperature ELSD (ELSD-LT; Shimadzu LC-2030 plus, Japan). Relative proportions of monomeric constituents determined for each MEL constituent were MEL-A (34.94%), MEL-B (28.46%), and MEL-C (11.32%), with the three monomers accounting for 74.72% of total MEL content by weight. At the same time, we carried out a simple verification of the detection method, which revealed that RSD values reflecting repeatability and stability were not >2.95%. In addition, the results of intra- and inter-day precision assessments (conducted on the same day and on 3 consecutive days, respectively) yielded RSD values that did not exceed 2.83%, thus demonstrating that the method was suitable for the quantitative detection of MELs (data not displayed).

### 2.3. Physicochemical Characterisation of MELs

To further confirm detailed structures of MELs produced by our *Candida antarctica* strain, high-resolution electrospray ionization mass spectrometry (HRESIMS) and nuclear magnetic resonance (NMR) analyses were employed. Mannosylerythritol lipid A (MEL-A) was obtained as a yellow oil with molecular formula C_34_H_60_O_13_ based on a HRESIMS peak at *m*/*z* 699.3925 [M + Na]^+^ calcd for C_34_H_60_O_13_Na, 699.3926 (Appendix A). As shown in Appendix A and Table 1, the resonance peaks of ^1^H NMR *δ*_H_ 3.66–3.99 m were assigned to H1-H4 of the erythritol moiety, while *δ*_H_ 5.50 dd, 5.06 dd, 5.24 t, 4.72 d, 3.69 m, 4.24 dd (H-6′b), 4.19 dd (H-6′a) corresponded to H1′-H6′ of mannose, respectively. Chemical shifts of δ_H_ 2.42 (m, -CH_2_C=O at the C-2′ position), 2.21 (m, -CH_2_C=O at the C-3′ position), 2.09 (s, -CH_3_C=O at the C-6′ position), 2.03 (s, -CH_3_C=O at the C-4′ position), and 1.31-1.23 (brs, -(-CH_2_-)_n_-), 0.86 (m, (-CH_3_)_2_) were also detected by ^1^H NMR. All 34 carbons were well resolved in the ^13^C NMR and via distortionless enhancement by polarization transfer (DEPT) spectra (Appendix A) and were classified as four methyl groups (*δ*_C_ 20.75, 20.68, 14.12, 14.11), nineteen methylene groups (*δ*_C_ 72.38, 63.60, 62.40, 34.16, 34.03, 31.88, 31.86, 29.48, 29.41, 29.33, 29.31, 29.28, 29.25, 29.09, 29.05, 25.05, 24.72, 22.68, 22.67), seven methine groups (*δ*_C_ 99.35, 72.55, 71.86, 71.19, 70.60, 68.56, 65.90), and four oxygenated quaternary carbons (*δ*_C_ 173.43, 172.73, 170.76, 169.46) within groups belonging to the MEL-A structure. As a result, HRESIMS and NMR results obtained in this work indicate successful the elucidation of the MEL-A (C8) chemical structure.

MEL-B was isolated as a yellow oil with a molecular formula of C_32_H_58_O_12_ according to HRESIMS, which indicated a value for *m*/*z* of 657.3835 for [M + Na]+ (calcd for C_32_H_58_O_12_Na, 657.3820, Appendix A). Constituent compounds of MEL-B were analyzed in detail via ^1^H NMR and ^13^C NMR (Appendix A and Table 1). Resonances of ^1^H NMR *δ*_H_ 3.77–3.56 m were assigned to H1-H4 of erythritol, while *δ*_H_ 5.50 dd, 4.91 dd, 4.77 m, 3.98 m, 3.85 m, 3.81 m (H-6′a) and 4.43 m (H-6′b) were assigned to H1′-H6′ of mannose. *δ*_H_ 2.13 (s, -CH_3_C=O at the C-6′ position), 1.64 (brs, -CO-CH_2_-CH_2_), 1.60 (brs, -CO-CH_2_-CH_2_), 1.26-1.24 (brs, -(CH_2_)_n_-), and 0.87 (m, (-CH_3_)_2_) were also inferred from ^1^H NMR spectral data. Crucial groups used to define MEL types corresponded to ^13^C NMR signals of *δ*_C_ 173.54, 173.4, 171.67 (three oxygenated quaternary carbons) and *δ*_C_ 20.86 (methyl of acetyl group), indicating that MEL-B was structurally distinct from MEL-A (four oxygenated quaternary carbons and two methyl groups within acetyl groups). Other ^13^C NMR signals included *δ*_C_ 99.39 (C-1′), 68.74 (C-2′), 71.23 (C-3′), 65.64 (C-4′), 74.58 (C-5′), 63.15 (C-6′), 63.63 (C-1), 73.12 (C-2), 71.93 (C-3), 72.23 (C-4), 22.61-31.94 (-(CH_2_)_n_-), 14.12 (-CH_3_), and 14.07 (CH_3_). Taken together, HRESIMS and NMR data confirmed the structure of MEL-B (C8).

MEL-C, obtained as a yellow oil, was determined to have a molecular formula C_30_H_54_O_12_ on the basis of HRESIMS data (*m*/*z* 629.3516 [M + Na]^+^, calcd for C_30_H_54_O_12_Na, 629.3507, Appendix A). As shown in Appendix A and Table 1, resonances of ^1^H NMR *δ*_H_ 3.54-3.73 m were assigned to H1-H4 of the erythritol moiety, while *δ*_H_ 5.50 dd, 5.10 dd, 5.18 t, 4.77 d, 4.04 m, 3.79 m (H-6′b), 3.77 m (H-6′a) corresponded to H1′-H6′ of mannose. *δ*_H_ 2.06 (s, -CH_3_C=O at the C-4′ position), 1.64 (brs, -CO-CH_2_-CH_2_), 1.54 (brs, -CO-CH_2_-CH_2_), 1.29-1.25 (brs, -(CH_2_)_n_-), and 0.88 (m, (-CH_3_)_2_) were also displayed from ^1^H NMR spectral data. All 30 carbons were well resolved in the ^13^C NMR and distortionless enhancement by polarization transfer (DEPT) spectra (Appendix A Appendix A) and were classified as three methyl groups (*δ*_C_ 20.73, 14.13, 14.07), seventeen methylene groups (*δ*_C_ 71.99, 63.48, 61.42, 34.11, 34.05, 31.90, 31.87, 29.51, 29.43, 29.35, 29.33, 29.11, 29.07, 25.06, 24.75, 22.69, 22.68), seven methine groups (*δ*_C_ 99.14, 74.88, 73.10, 71.10, 70.64, 68.70, 66.19), and three oxygenated quaternary carbons (*δ*_C_ 173.59, 172.79, 170.21) within groups belonging to the MEL-C structure. Therefore, HRESIMS and NMR results were elucidated as MEL-C (C7).

### 2.4. Effect of MEL and UVB on Cell Viability in HaCaT Cells

In this study, HaCaT cells were incubated with different concentrations of MEL-A, MEL-B, and MEL-C (0, 0.625, 1.25, 2.5, 5, 10, 20, 40, 80 μg/mL) to evaluate their effects on cell viability. As shown in Figure 2A–C, MEL-A, MEL-B, and MEL-C concentrations above 20 μg/mL were cytotoxic and led to a decrease in HaCaT cell viability. The viability of HaCaT cells exposed to 40 μg/mL MEL for 24 h was significantly decreased when compared to that of the control group, with survival rates of 10.00, 86.50, 48.00%, respectively. Based on these findings, 20 μg/mL was used in the subsequent experiments.

To establish a model of skin barrier damage, HaCaT cells were irradiated with different exposure intensities of UVB (0, 10, 20, 40, 80, 160, 320 mJ/cm^2^) to evaluate its degree of injury by measuring cell viability. The results indicated that exposure to 80 mJ/cm^2^ led to an 8.00% decrease in cell viability when compared to the control group (Figure 2D). Due to the fact that cell death might change the expressions of mRNA-levels genes and proteins, further biological studies were conducted at 40 mJ/cm^2^.

### 2.5. Effect of MELs on Contents of FLG and TGM1 in UVB-Induced HaCaT Cells

Purified MEL-A, MEL-B, and MEL-C preparations isolated from *Candida antarctica*-fermented broth were evaluated for their effects on the contents of key skin barrier proteins in UVB-induced HaCaT cells. ELISA results (Figure 3A) showed that, as compared with the control group, contents of TGM1 in the model group (HaCaT cells irradiated with UVB) decreased significantly, although no significant change in TGM1 content was observed as compared with that of the MEL-A-treated (20 μg/mL) experimental group. Meanwhile, in the MEL-B and MEL-C (20 μg/mL) experimental groups, the TGM1 content was significantly higher than that of the untreated model group, with a more marked MEL-B effect on TGM1 content than was observed for MEL-C (*p* < 0.0001). Next, we assessed changes in FLG content and obtained results that were similar to those obtained for TGM1 (Figure 3B). Therefore, MEL-B was selected for subsequent experiments to address our main research objective: to explore MELs’ targets and mechanisms underlying its protective effects against skin barrier damage.

### 2.6. Increased mRNA Expression of Genes Associated with Skin Barrier in UVB-Induced HaCaT Cells after MEL-B Treatment

We examined the mRNA-level expression of skin barrier-related genes encoding LOR, FLG, and TGM1 to assess the effects of MEL-B on the repair of skin barrier damage after UVB exposure. The results revealed that the mRNA-level expression of genes encoding LOR, and FLG was significantly decreased in UVB-irradiated HaCaT cells, while the expression levels of these mRNAs were significantly increased after being treated with MEL-B in a dose-dependent manner (Figure 4A,B). In addition, MEL-B treatment (20 μg/mL) boosted the mRNA-level expression of TGM1 genes that had decreased after UVB irradiation (*p* < 0.001), with the same effect observed in low concentrations of MEL-B doses (5 and 10 μg/mL, Figure 4C). Our results suggested that MEL-B could repair skin damage by restoring skin barrier function.

### 2.7. Upregulation of MEL-B Treatment on Expression Levels of LOR, FLG, TGM1 Protein in UVB-Induced HaCaT Cells

UVB irradiation (40 mJ/cm^2^) of HaCaT cells resulted in the inhibition of LOR, FLG, and TGM1 expression. As indicated in Figure 5A,B, expression levels of LOR protein in the model group decreased significantly as compared to the corresponding control group levels. In the experimental group treated with different doses of MEL-B (0, 2.5, 5, 10, 20 μg/mL), the expression of LOR protein increased in a dose-dependent manner with increasing MEL-B concentrations as compared with that of untreated model cells, with similar results obtained for the expression of TGM1 (Figure 5D). Furthermore, FLG protein expression levels also increased when compared with the model group due to the addition of MEL-B (Figure 5C). Accordingly, these data suggested that MEL-B effectively protected HaCaT cells from UVB-induced damage by upregulating the expression levels of LOR, FLG, and TGM1 proteins.

### 2.8. Protective Effects of MEL-B on Skin Barrier Damage as Assessed in SDS-Induced EpiKutis^®^3D Human Skin Cells

Histomorphological changes observed via H&E staining analysis demonstrated that in the SDS-treated (0.2%, *v*/*v*) EpiKutis^®^3D human skin cell damage model, the stratum corneum of the epidermal layer was loose and thickened, the living cell layer was damaged, vacuoles appeared, and obvious damage appeared as compared with the control group (Figure 6). There were reports in the literature that the expression of structural proteins of the upper spinous/granular layers (involucrin, profilaggrin–filaggrin, loricrin) increased following topical treatment with PPAR*α* activators. Furthermore, topically applied PPAR*α* activators also accelerated the recovery of barrier function following acute barrier abrogation [16]. WY14643 (50 μM) was used as the positive control drug in the experiment due to its supplementation, particularly with the PPAR*α* agonist WY14643, improving the homeostasis and barrier function of filaggrin (FLG)-deficient skin models by normalization of the free fatty acid profile [17]. As compared with the model group, in the positive control group, boundaries between the four layers of the model were clearer, living cell layer cells were arranged more compactly, and the stratum corneum integrity was significantly improved, with vacuoles and living cell layer damage both significantly reduced. In the MEL-B-treated experimental group, observed effects were similar to the corresponding effects observed in the positive control group, thus indicating that skin function was improved after MEL-B treatment (20 μg/mL).

### 2.9. Effects of MEL-B Treatment on Expression Levels of LOR, FLG, and TGM1 in Cells of the SDS-Induced EpiKutis^®^3D Human Skin Cell Damage Model

Using the SDS-induced EpiKutis^®^3D human skin cell damage model, expression levels of FLG, LOR, and TGM1 were measured via immunofluorescence (Figure 7A). The results indicated that expression levels of FLG, TGM1, and LOR in the negative control group were significantly lower than corresponding levels in model control group cells after SDS stimulation (0.2%, *v*/*v*). Meanwhile, levels of these proteins were significantly increased in the positive control group as compared to the corresponding levels observed in the model group and the MEL-B-treated (20 μg/mL) experimental group (Figure 7B). These data suggested that MEL-B protected SDS-induced EpiKutis^®^3D human skin cells from damage by upregulating the expression of FLG, LOR, and TGM1 proteins in vitro.

## 3. Discussion

The skin barrier, which is mainly composed of the epidermal layer, prevents loss of water and solutes, while maintaining homeostasis by serving as a basic protective barrier that prevents entry into the body of pathogens, irritants, and UV radiation [18]. At present, impaired skin barrier function and structural aberrations are thought to constitute the basis of numerous skin conditions and diseases, including atopic dermatitis (AD), eczema, psoriasis, ichthyosis, and so on [19,20,21].

Skin barrier dysfunction is associated with the reduced production of terminal differentiation-associated molecules, such as FLG and LOR, that are crosslinked by TGM1 to create a stable cornified protein shell known as the CE [22,23,24]. Results of a previous study revealed that *Lactobacillus rhamnosus* fermentation broth, when added to human epidermal skin model cells grown in vitro, improved epidermal barrier function by increasing the expression levels of LOR and FLG mRNAs [2]. Results of another study showed that expression levels of LOR and involucrin (IVL) in diseased skin associated with AD were also down-regulated as compared with healthy normal skin [25]. Similarly, results of this work showed that MEL-B effectively protected HaCaT cells from UVB-induced damage by up-regulating the intracellular expression of LOR, FLG, and TGM1 proteins. Meanwhile, we obtained similar results using an SDS-induced EpiKutis^®^3D human skin cell damage model after the treatment of cells with MEL-B (20 μg/mL). Taken together, these results thus indicated that the MEL-B induced expression of physical barrier-associated molecules that may have enhanced the function of skin as a physical barrier.

At present, mannosylerythritol lipids (MELs) were mainly obtained by *Candida antarctica* using vegetable oil, including soybean oil [26], olive oil [27,28], sunflower oil [29], castor oil [30] or glucose [31] as a substrate, or a mixture of the two in aerobic fermentation [32]. The single-batch fermentation yield of MELs can reach 165 g/L, for which the high purity product is usually separated by extraction and chromatographic chromatography [32]. They are generally categorized according to their degree of acetylation, as in MEL-A (di-acetylated at the C-4′ and C-6′ positions of mannose), MEL-B (mono-acetylated at the C-6′ position of mannose), and MEL-C (mono-acetylated at the C-4′ position of mannose). As recently adopted cosmetic ingredients that have attracted much attention, MEL-A and MEL-B are thought to possess great potential value as biosurfactants, due to their high hydrophobicities, good surface activities, liquid crystal-forming abilities, moisturizing activities, anti-oxidation activities, and hair repair effects [33,34]. In fact, several research groups studying MELs have shown their moisturizing properties to be similar to those of natural ceramides [33,35]. Meanwhile, other researchers have demonstrated the effectiveness of MELs in restoring damaged hair and improving hair smoothness and flexibility, while confirming the antioxidant properties of MELs for use as a potential anti-aging skin care component [36]. These results suggest that the monolayered structure of MELs facilitates their adsorption to damaged cells on skin surfaces, thus providing a convenient means of applying MELs to achieve a water-locking effect that aids the repair of damaged skin, warranting further study.

In this study, three mannosylerythritol lipids (MEL-A, MEL-B and MEL-C) were extracted and purified from *Candida antarctica* fermented olive oil culture broth using silica gel column chromatography followed by further purification via semipreparative HPLC. Ultimately, high-purity MEL preparations were obtained that were used to determine MELs chemical structures via NMR and HRESIMS. Next, we explored MEL-B effects on cells using two skin damage models (UVB-irradiated HaCaT cells and SDS-exposed EpiKutis^®^3D human skin cells) by assessing cellular FLG, LOR, and TGM1 mRNA genes and protein expression levels. Our results revealed that the levels of all three proteins were decreased in both skin damage models, as previously reported for both skin damage models. However, the MEL-B treatment of damaged cells of both models led to increased levels of these proteins. Furthermore, the results of the immunohistochemical staining revealed that the numbers of intracellular vacuoles and living cell layer damage were both significantly reduced as indicators of improved cell status after MEL-B treatment (20 μg/mL). Accordingly, MELs’ protection against UVB- or SDS-induced skin barrier damage depended on activities of MEL-B monomeric constituents as a key result that will be used to guide the development of future applications of MEL-B in the fields of cosmetics, environmental studies, and biotechnology.

## 4. Materials and Methods

### 4.1. Chemicals and Reagents

Column chromatography (CC) was performed using silica gel (60–80 mesh, 200–300 mesh, 300–400 mesh, Qingdao Haiyang Chemical Co. Ltd., Qingdao, China). Precoated silica gel 60 F (Merck, Darmstadt, Germany) was used for TLC analysis. Dulbecco’s Modified Eagle Medium (DMEM) and fetal bovine serum (FBS) were obtained from Thermo Fisher Scientific (Carlsbad, CA, USA) and Clark Bioscience (Claymont, DE, USA), respectively. Penicillin and streptomycin were purchased from Biosharp (Hefei, China). Radioimmunoprecipitation assay (RIPA) lysis buffer and the BCA protein assay kit were obtained from Beyotime Biotechnology (Jiangsu, China). Polyvinylidene fluoride (PVDF) membranes were obtained from GE Healthcare (Bensalem, PA, USA). Monoclonal primary antibodies against LOR, FLG, TGM1, GAPDH, p-p38, and p38 were purchased from Cell Signaling Technology (Beverly, CA, USA). All regular solvents and reagents were of reagent grade and were obtained from Sigma-Aldrich Chemical Co. (St. Louis, MO, USA), Acros Organics (Geel, Belgium), and J&K Scientific (Beijing, China).

### 4.2. General Experimental Procedures

#### 4.2.1. Production of MELs

A batch of properly stored, viable seed was inoculated into a 100 mL cone flask containing 10 mL YM medium (0.5% peptone, 1% glucose, 0.3% yeast extract, 0.3% malt extract) and cultured at 28 °C with shaking at 200 rpm for 24 h. Thereafter, 10 mL of the culture was inoculated into a 1000 mL Erlenmeyer flask containing 200 mL of sterile YM medium, then the flask was incubated under the abovementioned conditions to achieve a second expanded culture of seed-associated microbes. Next, 2.5 L of fermentation medium (0.1% yeast extract, 0.3% sodium nitrate, 0.03% magnesium sulfate, 0.03% potassium dihydrogen phosphate, 10% olive oil, and 0.01% defoaming agent) was transferred to a 5 L fermentation tank followed by autoclaving of the tank and medium. After the tank was allowed to cool to a temperature of 28 °C, 200 mL of the expanded seed microbial culture was added to the 5 L fermenter then the contents were incubated with stirring using a blade speed of 120 rpm at a temperature of 28 °C and an aeration rate of 1–1.5 vvm. During incubation, a small volume of culture was removed and assessed for yeast growth and contamination under magnification using a sampling microscope, with microbial assessment results and bacterial wet weight recorded every 24 h. At 48, 72, 96, and 120 h after culture initiation, a volume of olive oil that was 3, 6, 9, and 3% of the initial fermentation broth volume, respectively, was added to the culture. After 8 days (192 h) of incubation, fermentation was complete.

#### 4.2.2. Preliminary Separation and Purification of Fermentation Broth

After fermentation was complete, microbes in the broth were inactivated by heating to 80 °C for 2 h followed by centrifugation in a high-speed centrifuge at a temperature of 40 °C using a rotational speed of 8500 r/min for 30 min. Next, the centrifuged mixture was allowed to cool to room temperature then the supernatant was poured into another tube and the pellet was discarded. Thereafter, the supernatant was extracted with an equal volume of ethyl acetate. After the organic and water phases were separated, the water phase was extracted with organic reagents three times then the solvent was removed by vacuum distillation. To remove residual oil and fatty acids, the crude sticky MEL-containing extract that remained was washed twice with cyclohexane:methanol:water (1:6:3, *v*/*v*) that contained a solvent phase adjusted to a pH of 5.5 with 3 M hydrochloric acid and a water phase adjusted to a pH of 7.0. Next, spin distillation was conducted to remove the solvent under reduced pressure to obtain a crude product that was further purified to generate a purified mixture of MELs.

#### 4.2.3. Separation and Purification of MELs by Silica Gel Column Chromatography

Based on structural and polarity differences, MELs can be separated using silica gel column chromatography. After a glass chromatography column (3 × 40 cm) was washed and dried, it was filled with 300–400 mesh silica gel (Shandong Qingdao Marine Chemical Plant). Thereafter the MELs mixture (1.0 g) was dissolved in 5 mL of chloroform and loaded onto the column. Next, MEL-A was eluted in 900 mL of chloroform:acetone (5:1, *v*/*v*) then MEL-B was eluted in 800 mL of chloroform:acetone (3:1, *v*/*v*). Next, MEL-C was eluted in 600 mL of chloroform:acetone (2:1, *v*/*v*). The elution rate was set to 0.8 mL/min and a 20 mL volume of eluent was collected in each test tube. Purified MEL-A, MEL-B, and MEL-C fractions were used in subsequent experiments as detailed below.

Compounds were analyzed via thin-layer chromatography (TLC) using silica gel 60 G and a mobile phase of chloroform:methanol:ammonia (85:15:2, *v*/*v*). TLC plate development was stopped when the front of the solvent migrated 1.0 cm from the upper edge. Adequate migration of compounds was preliminarily verified by exposure of TLC plates to iodine vapor to induce a color change. Isolated compounds on the TLC plates were located by charring at 105 °C for 5 min after plates had been evenly sprayed with chromogenic agent (100 mg anthrone, 2 mL concentrated sulfuric acid, 18 mL anhydrous ethanol) until blue spots appeared.

#### 4.2.4. Quantification of MELs by High-Performance Liquid Chromatography (HPLC)

Quantification of MELs was carried out via high-performance liquid chromatography (HPLC) on a Kromasil C_18_ column (i.d. 4.6 × 250 mm, 5 μm, Sweden) using an HPLC system equipped with a low-temperature evaporative light-scattering detector (ELSD-LT; Shimadzu LC-2030 Plus equipped with quaternary gradient pump, ELSD and UV detector, Japan). For HPLC, a gradient solvent program was used that was based on various ratios of water to acetonitrile at a flow rate of 1.0 mL/min and a sample injection volume of 10 μL. A stepwise elution gradient was employed using solvent A (water) and solvent B (acetonitrile) and maintained at 30 °C. The gradient elution program was as follows: 80% B for 0–10 min, 80–90% B for 10–15 min, 90% B for 15–40 min. ELSD detector settings included a gain of 6, drift tube temperature of 40 °C, and pressure of 358 kpa.

#### 4.2.5. Structure Analysis of MELs by HRESIMS and NMR

HRESIMS data were collected and recorded using an Agilent 6540 Series quadrupole time-of-flight (Q-TOF) liquid chromatography/mass spectrometry (LC/MS) system (Agilent Technologies, Santa Clara, CA, USA). Q-TOF LC/MS settings included positive ion mode, ion source voltage 3.5 kV, capillary voltage 135 V, taper hole voltage 65 V, drying gas temperature 350 °C, drying gas flow rate 8 L/min, atomization pressure 30 psi, full scanning range 100 to 1700 *m*/*z*, mobile phase water A:acetonitrile B (1:1, *v*/*v*), and a flow rate 0.3 mL/min using a Kromasil C_18_ chromatographic column (i.d. 2.1 × 50 mm, 2.5 μm, Sweden).

Refined glycolipid (10.0 mg) was dissolved in 0.5 mL of CDCl_3_ (99.9%) then NMR experiments were conducted using a Bruker AVANCE III 500 and 600 MHz spectrometer (Bruker Inc., Rheinstetten, Germany) based on an internal standard consisting of residual CDCl_3_ containing 0.03% *v*/*v* trimethylsilane (TMS). The NMR system was equipped with a 5-mm-diameter BBO probe and was operated using pulse sequence settings as follows: zg30, spectral width (SW) 22.027 ppm, sampling points (TD) 65536, scanning times (NS) 16, relaxation time (D1) 1 s, empty scanning times (DS) 2, detection temperature 25 °C, coaxial NMR sample tube.

Mannosylerythritol lipid A (MEL-A) ^1^H NMR (500 MHz, CDCl_3_) *δ* 5.50 (d, *J* = 3.2 Hz, H-1′), 5.24 (t, *J* = 10.0 Hz, H-3′), 5.06 (dd, *J* = 10.1, 3.3 Hz, H-2′), 4.72 (d, *J* = 6.2 Hz, H-4′), 4.24 (dd, *J* = 12.2, 5.8 Hz, H-6′b), 4.19 (dd, *J* = 12.2, 2.6 Hz, H-6′a), 3.99 (dd, *J* = 10.6, 3.4 Hz, H-4b), 3.84 (dd, *J* = 10.6, 5.9 Hz, H-4a), 3.76 (m, H-1), 3.72 (m, H-3), 3.69 (m, H-5′), 3.66 (m, H-2), 2.42 (m, -CH_2_C=O at the C-2′ position), 2.21 (m, -CH_2_C=O at the C-3′ position), 2.09 (s, -CH_3_C=O at the C-6′ position), 2.03 (s, CH_3_C=O at the C-4′ position), 1.64 (brs, -CO-CH_2_-CH_2_), 1.54 (brs, -CO-CH_2_-CH_2_), 1.31-1.23 (brs, (-CH_2_)_n_-), 0.86 (m, (-CH_3_)_2_); ^13^C NMR (126 MHz, CDCl_3_) *δ* 173.43 (C=O), 172.73 (C=O), 170.76 (C=O), 169.46 (C=O), 99.35 (CH), 72.55 (CH), 72.38 (CH_2_), 71.86 (CH), 71.19 (CH), 70.60 (CH), 68.56 (CH), 65.90 (CH), 63.60 (CH_2_), 62.40 (CH_2_), 34.16 (CH_2_), 34.03 (CH_2_), 31.88 (CH_2_), 31.86 (CH_2_), 29.48 (CH_2_), 29.41 (CH_2_), 29.33 (CH_2_), 29.31 (CH_2_), 29.28 (CH_2_), 29.25 (CH_2_), 29.09 (CH_2_), 29.05 (CH_2_), 25.05 (CH_2_), 24.72 (CH_2_), 22.68 (CH_2_), 22.67 (CH_2_), 20.75 (CH_3_), 20.68 (CH_3_), 14.12 (CH_3_), 14.11 (CH_3_).

Mannosylerythritol lipid B (MEL-B) ^1^H NMR (500 MHz, CDCl_3_) *δ* 5.50 (dd, *J* = 6.8, 3.8 Hz, H-1′), 4.91 (dd, *J* = 9.9, 3.0 Hz, H-2′), 4.77 (d, *J* = 5.6 Hz, H-3′), 4.43 (dd, *J* = 5.4, 3.9 Hz, H-6′b), 3.98 (m, H-4′), 3.85 (m, H-5′), 3.81 (m, H-6′a), 3.77 (dd, *J* = 5.1, 1.6 Hz, H-3), 3.75 (dd, *J* = 5.1, 2.9 Hz, H-4b), 3.72 (m, H-1), 3.66 (m, H-2), 3.56 (m, H-4a), 2.40 (m, -CH_2_C=O at the C-2′ position), 2.29 (m, -CH_2_C=O at the C-3′position), 2.13 (s, -CH_3_C=O at the C-6′ position), 1.64 (brs, -CO-CH_2_-CH_2_), 1.60 (brs, -CO-CH_2_-CH_2_), 1.26-1.24 (brs, -(CH_2_)_n_-), 0.87 (m, (-CH_3_)_2_); ^13^C NMR (126 MHz, CDCl_3_) *δ* 173.54 (C=O), 173.41 (C=O), 171.67 (C=O), 99.39 (CH), 74.58 (CH), 73.12 (CH), 72.23 (CH_2_), 71.93 (CH), 71.23 (CH), 68.74 (CH), 65.64 (CH), 63.63 (CH_2_), 63.15 (CH_2_), 34.20 (CH_2_), 34.11 (CH_2_), 31.94 (CH_2_), 31.88 (CH_2_), 29.51 (CH_2_), 29.43 (CH_2_), 29.38 (CH_2_), 29.35 (CH_2_), 29.32 (CH_2_), 29.28 (CH_2_), 29.11 (CH_2_), 29.08 (CH_2_), 25.08 (CH_2_), 24.72 (CH_2_), 22.68 (CH_2_), 22.61 (CH_2_), 20.86 (CH_3_), 14.12 (CH_3_), 14.07 (CH_3_).

Mannosylerythritol lipid C (MEL-C) ^1^H NMR (600 MHz, CDCl_3_) *δ* 5.50 (dd, *J* = 10.6, 2.9 Hz, H-1′), 5.18 (t, *J* = 9.9 Hz, H-3′), 5.10 (dd, *J* = 10.1, 3.2 Hz, H-2′), 4.77 (d, H-4′), 4.04 (m, H-5′), 3.79 (m, H-6′b), 3.77 (m, H-6′a), 3.75 (m, H-3), 3.73 (m, H-1), 3.68 (m, H-4a), 3.65 (m, H-2), 3.54 (m, H-4b), 2.43 (m, -CH_2_C=O at the C-2′ position), 2.22 (m, -CH_2_C=O at the C-3′position), 2.06 (s, CH_3_C=O at the C-4′ position), 1.64 (m, -CO-CH_2_-CH_2_), 1.54 (m, -CO-CH_2_-CH_2_), 1.29-1.25 (brs, -(CH_2_)_n_-), 0.88 (m, (-CH_3_)_2_); ^13^C NMR (151 MHz, CDCl_3_) *δ* 173.59 (C=O), 172.79 (C=O), 170.21 (C=O), 99.14 (CH), 74.88 (CH), 73.10 (CH), 71.99 (CH_2_), 71.10 (CH), 70.64 (CH), 68.70 (CH), 66.19 (CH), 63.48 (CH_2_), 61.42 (CH_2_), 34.11 (CH_2_), 34.05 (CH_2_), 31.90 (CH_2_), 31.87 (CH_2_), 29.51 (CH_2_), 29.43 (CH_2_), 29.35 (CH_2_), 29.33 (CH_2_), 29.11 (CH_2_), 29.07 (CH_2_), 25.06 (CH_2_), 24.75 (CH_2_), 22.69 (CH_2_), 22.68 (CH_2_), 20.73 (CH_3_), 14.13 (CH_3_), 14.07 (CH_3_).

### 4.3. Bioactivity Assay

#### 4.3.1. Cell Culture

Human epidermal keratinocyte (HaCaT) cells (Shanghai Fuheng Biotechnology Co., Ltd. Shanghai, China) were cultured in 10% DMEM, 10% FBS, 100 units/mL penicillin, and 100 μg/mL streptomycin in an incubator containing a humidified CO_2_:air (5:95, *v*/*v*) atmosphere at 37 °C. When cell growth reached the logarithmic growth phase, the cells were transferred to 35 mm diameter cell culture dishes then were used for experiments when they reached 80% confluence. Prior to UVB irradiation, the culture medium was replaced with 1 mL of phosphate-buffered saline (PBS, Gibco Life Technologies, New York, NY, USA) per well.

#### 4.3.2. UVB Irradiation and Treatments

HaCaT keratinocytes (80% confluent) in uncovered 35 mm diameter cell culture dishes at room temperature were irradiated (40 mJ/cm^2^ UVB irradiation) using a UV crosslinker system (Hoefer Scientific Instruments, San Francisco, CA, USA) with an emission wavelength peak of 312 nm. Next, irradiated cells were treated with MEL-B (0, 2.5, 5, 10, or 20 μg/mL) and cultured for 24 h.

#### 4.3.3. Cell Viability Assay

The Hacat cells were seeded in 96-well plates (8 × 10^3^ cells/well), followed by 100 μL DMEM medium added to each well. When the cells growth reached 80%, they were treated with MEL-A, MEL-B, and MEL-C at various concentrations of 0, 0.625, 1.25, 2.5, 5, 10, 20, 40, and 80 μg/mL or UVB irradiated at 10, 20, 40, 80, 160, and 320 mJ/cm^2^. The wells with no drug added at the same incubation time were used as the control group. After the treatment for 24 h, MTT [3-(4, 5-dimethyl-thiazol-2yl)-2, 5-diphenyl tetrazolium bromide] (0.5 mg/mL, Sigma-Aldrich, Shanghai branch, China) solution was added to each well, and kept at 37 °C for 4 h to determine cell viability according to the instructions provided with the kit. After removing the supernatant, DMSO (150 μL) was added into dissolved formazan crystals for the measurement of the absorbance at 490 nm using a microplate reader (TECAN A-5082, Magellan, Austria). The data are presented as the percentage of viable cells relative to the vehicle control.

#### 4.3.4. ELISA

HaCaT cells were incubated with concentrations of MEL-A, MEL-B, MEL-C (20 μg/mL), and irradiated with exposure intensities of UVB (40 mJ/cm^2^). The wells with no drug added at the same incubation time were used as the control group. Levels of the proteins TGM1 and FLG were determined using commercial ELISA kits (Boster Biological Technology Co., Ltd., Wuhan, Hubei, China) according to the manufacturer’s protocols.

#### 4.3.5. Quantitative Real-Time PCR (RT-PCR) Analysis

Total RNA was extracted using TRIzol Reagent (TIANGEN, Beijing, China). RNA purity and integrity were assessed spectroscopically using a NanoDrop 2000/c spectrophotometer (ThermoScientific, Waltham, MA, USA). 2 μg of total RNA was reverse transcribed into cDNA using a PrimeScript RT reagent Kit (TaKaRa, Dalian, China). RT-PCR was performed using a CFX96 Real-Time PCR Detection System (Bio-Rad, Hercules, CA, USA). The program is as follows: 95 °C 5 min, 95 °C 15 s, 60 °C 30 s, 72 °C 30 s for 40 cycles. The 2^−ΔΔCt^ value method was used for data analysis, with results normalized to GAPDH expression levels. The primer sequences are listed in Table 2.

#### 4.3.6. Western Blotting

After cells were lysed in RIPA buffer supplemented with protease inhibitors with a phosphatase inhibitor cocktail, a BCA assay was used to measure protein concentration. Next, equal amounts of protein (50 μg/sample) were electrophoresed on 10% SDS-PAGE gels, then the proteins were electroblotted onto PVDF membranes. After PVDF membranes were blocked for 2 h in 5% BSA, they were probed overnight with appropriate primary antibodies (1:1000) at 4 °C. Next, membranes were washed three times with Tris-buffered saline containing Tween (TBST) and incubated with appropriate HRP-conjugated secondary antibodies (1:5000), then they were washed and developed using Enhanced Chemiluminescence (ECL) reagent as per the manufacturer’s instructions. Expression levels of proteins were normalized based on GAPDH levels. A FluorChem HD2 system (ProteinSimple, San Jose, CA, USA) was used for image acquisition and analysis.

#### 4.3.7. Immunohistochemistry

EpiKutis^®^3D cell-based model was provided by Guangdong BioCell Biotechnology Co., Ltd., and experiments were conducted using a blank control group, model group (0.2% SDS), positive control group (0.2% SDS and 50 μM WY14643), and experimental group (0.2% SDS and 20 μg/mL MEL-B). The cells of all four of the abovementioned groups were cultured in an incubator with 5% CO_2_ at 37 °C for 24 h then were fixed for 30 min in paraformaldehyde of 40 g/L. After fixed samples were washed with PBS, samples were dehydrated by stepwise immersion in solutions of increasing ethanol concentration, embedded in paraffin overnight, then were sealed with dry gum and sliced into 8-mm-thick sections. Sections were stained with hematoxylin and eosin (H&E) then the cell morphology was assessed microscopically using an M8 microscope (PreciPoint GmbH, Freising, Germany).

#### 4.3.8. Immunofluorescence

EpiKutis^®^3D human skin cells were cut into 8-mm-thick sections, deparaffinized, rehydrated, and sectioned as described above in Section 4.3.7. Sections were incubated in endogenous peroxidase blocking solution for 10 min then non-specific binding of proteins was blocked by the immersion of sections in normal non-immune serum for 10 min followed by the incubation of sections with primary antibodies for 1 h. Thereafter, the sections were treated with biotin-conjugated secondary antibody and freshly prepared 3,3′ diaminobenzidine (DAB) solution (Santa Clara, CA, USA) then were counterstained with hematoxylin. Finally, immunostained sections were observed under magnification using a confocal microscope (C2, Nikon, Tokyo, Japan).

#### 4.3.9. Statistical Analysis

Data are expressed as means ± SD of three independent experiments. A one-way analysis of variance (ANOVA) and the Tukey’s test were used to assess the significance of differences, with statistical analyses conducted using GraphPad Prism 7.0 (GraphPad Software, San Diego, CA, USA). *p* values < 0.05 was considered statistically significant for all experiments.

## 5. Conclusions

Herein, our results indicated that MEL-B protected both HaCaT and EpiKutis^®^3D skin cell models from UVB- and SDS-induced damage by up-regulating the expression of the skin barrier damage-associated key mRNA genes and proteins LOR, FLG, and TGM1. Thus, these results provide evidence supporting beneficial MEL-B effects against skin damage and support its application as a cosmetic additive for use in preventing skin barrier damage. Further mechanistic studies are underway that should soon shed light on the mechanisms and targets associated with MEL-B’s skin-protective effects.

## Figures and Tables

**Figure 1 molecules-27-04648-f001:**
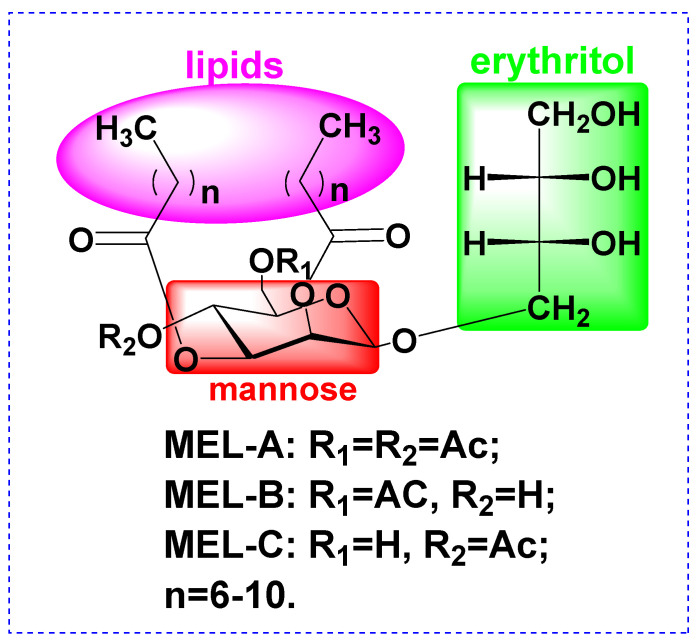
Structure of Mannosylerythritol lipids (MELs).

**Figure 2 molecules-27-04648-f002:**
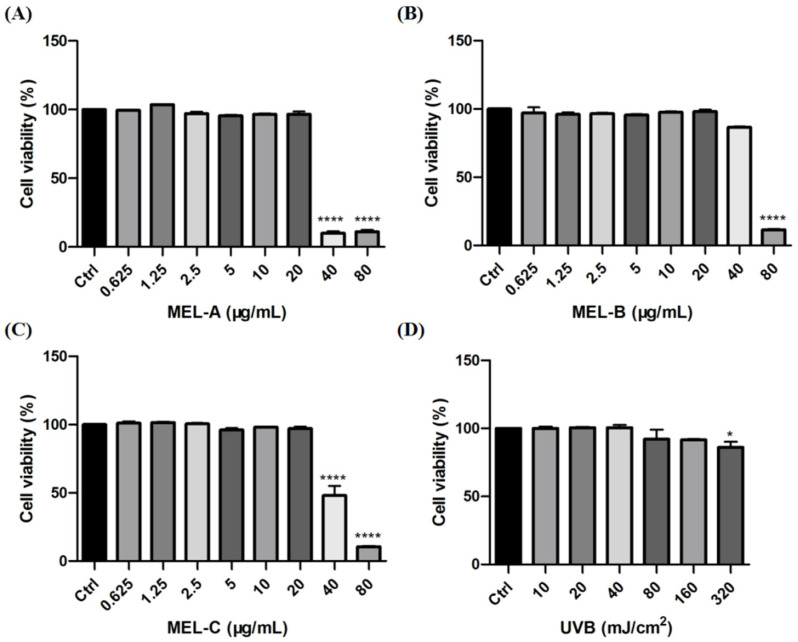
Effects of MEL and UVB on cell viability in HaCaT cells. (**A**) HaCaT cell viability was assayed by an MTT assay after various concentrations of MEL-A incubation for 0, 0.625, 1.25, 2.5, 5, 10, 20, 40 or 80 μg/mL. (**B**,**C**) HaCaT cells were treated with different doses of MEL-B and MEL-C for 24 h and their cell viability. (**D**) Effect of different exposure intensities of UVB on cell viability in HaCaT cells. Values are expressed as means ± SD (*n* = 3). * *p* < 0.05, **** *p* < 0.0001 vs. control group (Ctrl).

**Figure 3 molecules-27-04648-f003:**
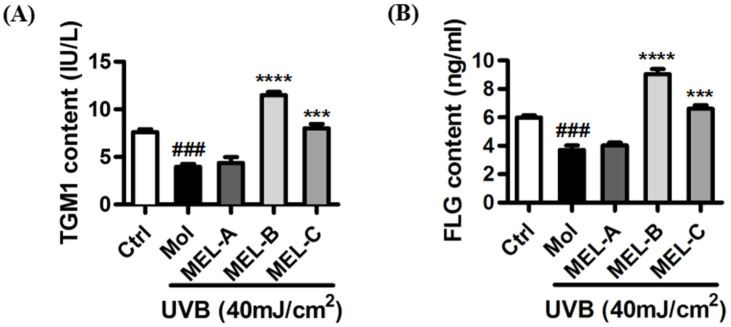
Effects of MEL monomeric constituents on the content of TGM1 and FLG proteins in UVB-irradiated HaCaT cells. (**A**) After UVB irradiating, the content of TGM1 in the model group decreased significantly (*p* < 0.001) when compared with the control group. MEL-B and MEL-C (20 μg/mL) significantly increased the contents of TGM1 when compared with the model group. (**B**) FLG content was significantly higher than that of the untreated model group, with a more marked MEL-B effect on FLG content than was observed for MEL-C (*p* < 0.0001). Values are expressed as means ± SD (*n* = 3). ### *p* < 0.001 vs. the control group (Ctrl); *** *p* < 0.001, **** *p* < 0.0001 vs. the UVB-irradiated group (Mol).

**Figure 4 molecules-27-04648-f004:**
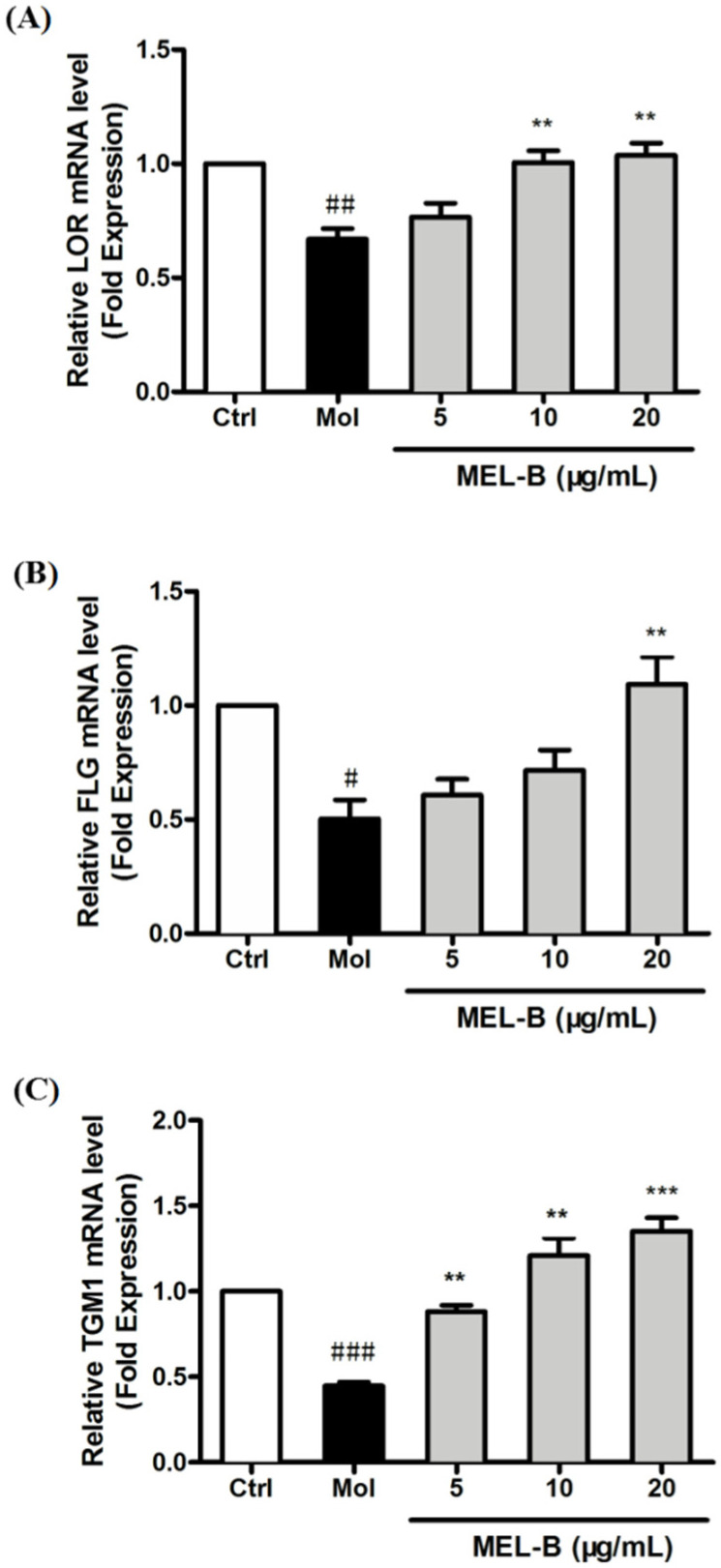
Different concentrations of MEL-B treatment increased the mRNA-level expression of genes associated with the skin barrier in UVB-irradiated HaCaT cells. mRNA levels reflecting the expression of genes encoding LOR (**A**), FLG (**B**), and TGM1 (**C**) were quantified using RT-PCR after the normalization of results to GAPDH mRNA expression levels (loading control). Increased mRNA expression of LOR, FLG, and TGM1 genes in UVB-induced HaCaT cells after MEL-B treatment (20 μg/mL, *p* < 0.01). Values are expressed as means ± SD (n = 3). # *p* < 0.05, ## *p* < 0.01, ### *p* < 0.001 vs. the control group (Ctrl); ** *p* < 0.01, *** *p* < 0.001 vs. the UVB-irradiated group (Mol).

**Figure 5 molecules-27-04648-f005:**
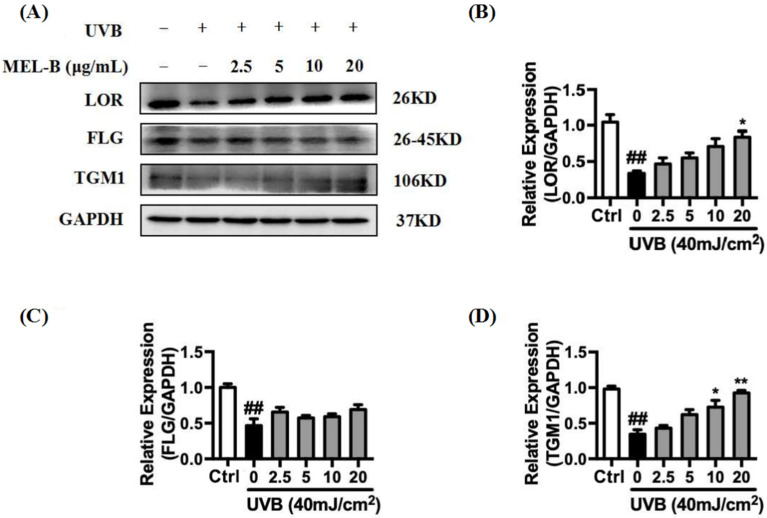
Effects of MEL-B treatment on UVB-irradiated HaCaT cells. (**A**) HaCaT cells were irradiated with UVB (40 mJ/cm^2^) then treated with MEL-B (0, 2.5, 5, 10, 20 μg/mL) followed by 24-h culture. Relative expression levels of LOR, FLG, and TGM1 were decreased by UVB radiation. (**B**) Relative expression levels of LOR were increased after MEL-B incubation in dose dependent. (**C**) Upregulation of MEL-B treatment on expression levels of FLG protein. (**D**) Dose-dependent MEL-B effect on upregulation of TGM1 protein expression in UVB-irradiated HaCaT cells. GAPDH antibody was used as loading control. Values are expressed as means ± SD (*n* = 3). ## *p* < 0.01 vs. the control group (Ctrl); * *p* < 0.05, ** *p* < 0.01 vs. the UVB-irradiated (model) group.

**Figure 6 molecules-27-04648-f006:**
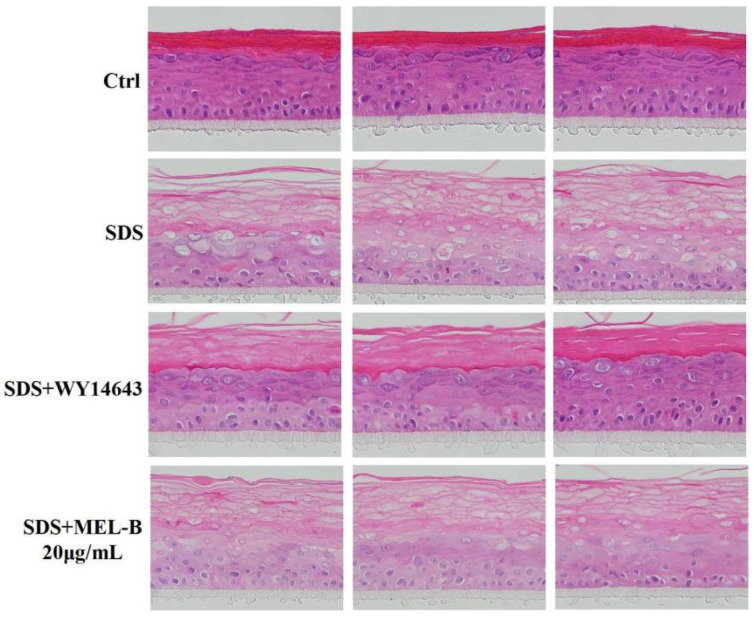
Protective effect of MEL-B on the SDS-induced EpiKutis^®^3D human skin cell damage model as assessed via H&E staining, with red-stained areas corresponding to keratin and blue–purple-stained areas corresponding to nucleic acids. WY14643 (50 μM) was used as the positive control drug in the experiment. Numbers of vacuoles and damage of the living cell layer were both significantly reduced, indicating that impaired skin function due to SDS exposure was markedly reversed after MEL-B treatment (20 μg/mL).

**Figure 7 molecules-27-04648-f007:**
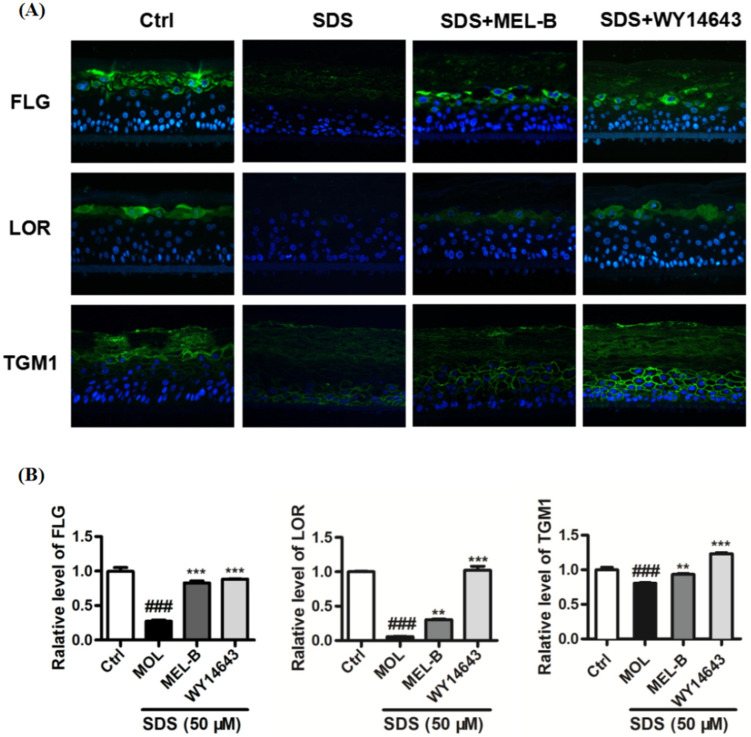
Effects of MEL-B treatment on SDS-induced EpiKutis^®^3D human skin cells as assessed via immunofluorescence staining, with blue-stained areas corresponding to nucleus and yellow–green-stained areas corresponding to cell membrane. WY14643 (50 μM) was used as the positive control drug in the experiment. (**A**) Expression of key proteins were analyzed and treated with MEL-B (20 μg/mL) for 24 h in EpiKutis^®^3D human skin cells by a confocal microscope. Scale bar = 100 μm. (**B**) Relative expression levels of FLG, LOR, TGM1 were significantly increased after MEL-B incubation as compared to the corresponding levels observed in the SDS-induced group (Mol). Values are expressed as means ± SD (*n* = 3). ### *p* < 0.001 vs. the control group (Ctrl); ** *p* < 0.01, *** *p* < 0.001 vs. the SDS-induced (Mol) group.

**Table 1 molecules-27-04648-t001:** ^1^H and ^13^C NMR Spectroscopic Data of MELs (*δ* in ppm, *J* in Hz).

	MEL-A ^a^		MEL-B ^a^		MEL-C ^b^	
Functional Groups	^1^H NMR *δ* (ppm)	^13^C NMR *δ* (ppm)	^1^H NMR *δ* (ppm)	^13^C NMR *δ* (ppm)	^1^H NMR *δ* (ppm)	^13^C NMR *δ* (ppm)
D-Mannose						
H-1′ (C-1′)	5.50, d	99.35	5.50, dd	99.39	5.50, dd	99.14
H-2′ (C-2′)	5.06, dd	68.56	4.91, dd	68.74	5.10, dd	68.70
H-3′ (C-3′)	5.24, t	70.60	4.77, m	71.23	5.18, t	70.64
H-4′ (C-4′)	4.72, d	65.90	3.98, m	65.64	4.77, d	66.19
H-5′ (C-5′)	3.69, m	72.55	3.85, m	74.58	4.04, m	74.88
H-6′ (C-6′) a	4.19, dd	62.40	3.81, m	63.15	3.77, m	61.42
H-6′ (C-6′) b	4.24, dd	62.40	4.43, m	63.15	3.79, m	61.42
*meso*-Erythritol						
H-1 (C-1)	3.76, m	63.60	3.72, m	63.63	3.73, m	63.48
H-2 (C-2)	3.66, m	71.86	3.66, m	73.12	3.65, m	73.10
H-3 (C-3)	3.72, m	71.19	3.77, m	71.93	3.75, m	71.10
H-4a (C-4a)	3.84, dd	72.38	3.56, m	72.23	3.68, m	71.99
H-4b (C-4b)	3.99, dd	72.38	3.75, m	72.23	3.54, m	71.99
Acetyl group(s)						
-CH_3_	2.09, 2.03, s	(20.75, 20.68)	2.13, s	20.86	2.06, s	20.73
-C=O		(170.76, 169.46)		171.67		170.21
Acyl groups						
-CH_3_	0.86, m	(14.12, 14.11)	0.87, m	(14.12, 14.07)	0.88, m	(14.13, 14.07)
-(CH_2_)_n_-	1.23–1.31, brs	(22.67–31.88)	1.24–1.26, brs	(22.61–31.94)	1.25–1.29, brs	(22.68–31.90)
-CO-CH_2_- (C-2′) position	2.42, m	34.16	2.40, m	34.11	2.43, m	34.11
-CO-CH_2_- (C-3′) position	2.21, m	34.03	2.29, m	34.20	2.22, m	34.05
-C=O		(173.43, 172.73)		(173.54, 172.41)		(173.59, 172.79)

s, singlet; d, doublet; dd, double doublet; t, triplet; m, multiplet; brs, broad singlet. ^a^ Data were measured at 500 MHz for ^1^H and 126 MHz for ^13^C NMR in CDCl_3_; ^b^ Data were measured at 600 MHz for ^1^H and 151 MHz for ^13^C NMR in CDCl_3._

**Table 2 molecules-27-04648-t002:** Primer sequences used for quantification of gene expression.

Gene	Sequence
LOR-F	5′-GAGCTACGGAGGCGTCTCTA-3′
LOR-R	5′-AGAGTAGCCGCAGACAGA-3′
FLG-F	5′-GGTCTGGACGTTCAGGGTCT-3′
FLG-R	5′-GGATGTGGTGTGGCTGTGAT-3′
TGM1-F	5′-CCCCAAGAGACTAGCAGTGG-3′
TGM1-R	5′-AGACCAGGCCATTCTTGATG-3′
GAPDH-F	5′-GTGAAGGTCGGAGTCAACG-3′
GAPDH-R	5′-TGAGGTCAATGAAGGGGTC-3′

F, Forward; R, Reverse.

## Data Availability

The data presented in this study are available in this article. Other data that support the findings of this study are available upon request from the corresponding author.

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
