# Peer review of "Screening and Research on Skin Barrier Damage Protective Efficacy of Different Mannosylerythritol Lipids"

_molecules, 2022, doi:10.3390/molecules27144648_

Round 1
Reviewer 1 Report
The paper entitled: “Screening and Research on Skin Barrier Damage Protective Efficacy of Different Mannosylerythritol Lipids” is well-written, well-designed and interesting.
It needs minor revisions.
My comments are:
- The effects of MEL monomeric constituents on content of TGM1 and FLG proteins in UVB-irradiated HaCaT cells are reported in the Figure 4 of the manuscript. Please specify, in the “Materials and Methods” section and in the caption to the Figure 4, the concentration of MEL-A, MEL-B, and MEL-B used for HaCaT cell treatment.
- The effect of MEL-A, MEL-B, and MEL-C on HaCaT cell viability (after 24-h of incubation) should be evaluated and reported in the manuscript in order to better understand the different effect of MEL-A, MEL-B, and MEL-C on the content of TGM1 and FLG proteins in UVB-irradiated HaCaT cells.
- The time of irradiation with UVB (at emission wavelength peak of 312 nm) and the effect of irradiation on HaCaT cell viability should be also reported.
- Please indicate control cells in the same way in all Figures, for example as Ctrl instead of CON (as indicated in Figures 4 and 5).
- The use of the compound WY14643 (positive control group of Figures 6 and 7) should be justified in the text of the manuscript.
- The different treatment groups should be better described in the captions to all figures.
- Page 6, line 161: please change “MEL” to “MELs”.
- Page 11, line 361: please change “MEL” to “MELs”.
- Page 12, line 373: please change “MEL” to “MELs”.
- The indication of Figures (page 9, lines 256 and 257) should be deleted from the text of the discussion.
Author Response
We appreciate the constructive and positive comments and suggestions from reviewers and have addressed all concerns and comments point-by-point and made relevant changes to the manuscript accordingly. Please check the attachment.

Reviewer 2 Report
Please present explicity the structure of the compounds MEL-A, B, and C. Figure 1 is not clear enough to understand the chemical structure of the molecules.
The NMR details of the peak assignment and the mass spectrometry description may be summarized and details described as supplementary materials.
Since the structures are not clearly shown, it is hard to make conclusions about the structure-function relationships between the three mannosylerythritol lipids. there is so little information in the literature about those molecules that is well deserved to explain more about the possible role of substituents and other possibilities of applications.
Why MEL-B only is used for explaining the skin damage in the EpiKutis3D model system?
Also, please comments something about the production and biotechnology of those compounds obtained from Candida antartica fermentation in oil, yields and the downstream processes.
Author Response
We appreciate the constructive and positive comments and suggestions from reviewers and have addressed all concerns and comments point-by-point and made relevant changes to the manuscript accordingly. Please check the attachment

Reviewer 3 Report
This manuscript describes an interesting study of the bioactivity assessment of mannosylerythritol lipids. The authors extracted three MELs from Candida antarctica cultures and purified using silica gel-based column chromatography and semipreparative HPLC. The results showed that MELs provide protective effects against skin damage by and protect HaCaT cells from UVB-induced damage. Histopathological analysis revealed markedly reduced intracellular vacuolation and cell damage reflecting improved skin function after treatment with MEL-B. The work is interesting and carefully prerformed, however, there are a few points which need to be addressed by the authors:
1) Figure 2 and Figure 3 are provided with low resolution. The authors should replace with improved versions.
2) The authors studied the expression of LOR, FLG, and TGM1 using western blot. Did they consider using RT-PCR to further confirm their results ?
3) Why do authors studied the expression of LOR, FLG, and TGM1 ? Did they consider the expression of other important proteins ?
Author Response

(The authors gave the same response as above.)

Round 2
Reviewer 3 Report
The manuscript was suitably revised and the authors responded to my concerns. It can be accepted.